# Influence of Ozone on the Biochemical Composition of Birch Sap

**Simona Paulikienė** [1],*, **Justas Mingaila** [2], **Vladas Vilimas** [1], **Edmundas Bartkevičius** [2], **Pranas Viskelis** [3] and **Algirdas Raila** [1]

1. Institute of Energy and Biotechnology Engineering, Agriculture Academy, Vytautas Magnus University, K. Donelaičio str. 58, 44248 Kaunas, Lithuania; vladas.vilimas@vdu.lt (V.V.); algirdas.raila@vdu.lt (A.R.)
2. Institute of Forest Biology and Silviculture, Agriculture Academy, Vytautas Magnus University, K. Donelaičio str. 58, 44248 Kaunas, Lithuania; justasmingaila@gmail.com (J.M.); edmundas.bartkevicius@vdu.lt (E.B.)
3. Institute of Horticulture, Lithuanian Research Centre for Agriculture and Forestry, Kaunas str. 30, LT-54333 Babtai, Kaunas District, Lithuania; pranas.viskelis@lammc.lt
* Correspondence: simona.paulikiene1@vdu.lt

**Abstract:** Studies have shown that ozone is a good oxidizer and a strong disinfectant. There are many uses for ozone in the food industry, but there is relatively little information about the influence of ozone on biochemical composition and the capacity to reduce the number of microorganisms in birch sap. In this study, sap was ozonated at different intervals for 5 min ($O_3$: $0.087 \pm 0.009$ mg $L^{-1}$), 10 min, 15 min, 20 min, 25 min, or 30 min ($O_3$: $0.99 \pm 0.09$ mg $L^{-1}$). The parameters of the birch sap were studied immediately after the ozone treatment as well as during storage for seven days at 2 °C and for five days at 20 °C. The parameters of ozonated birch sap were compared with the parameters of fresh sap (control). The microbiological analysis included total bacterial count, lactic acid bacterial count, and yeast and mold count. Birch sap color, pH, titratable acidity, and °Brix values were also determined. Evaluation of monosaccharides, sucrose, total sugars, and ascorbic acid was carried out in fresh sap as well as sap ozonated for 30 min, immediately after ozonation. The results show the statistical significance of the inactivation of microorganisms after treatment in most cases. The microorganism counts gradually reduced with increasing intervals of ozone treatment. The best results were obtained after 25 and 30 min of ozonation. Ozone treatment did not significantly influence the pH, titratable acidity, or °Brix statistically. Values of monosaccharides, sucrose, total sugars, and ascorbic acid were influenced within the margin of error. Ozone had a significant influence on the chroma and hue angle.

**Keywords:** *Betula pendula*; biochemical composition; birch sap; ozone treatment; microbiological contamination; quality

## 1. Introduction

Birch sap is a forest resource with a deep tradition of use in northern and eastern Europe, where it is obtained from the silver birch (*Betula pendula* Roth) and the curly birch (*Betula pubescens* Ehrh) [1], as well as in North America, where it is obtained from the white birch (*Betula papyrifera* Marsh), sweet birch (*Betula lenta* L.), and yellow birch (*Betula alleghaniensis* Britt.) [2]. Birch sap consists mainly of water and a solution of sugars, which contains various amounts of amino and organic acids, peptides, vitamins, and phenolic substances [3–7]. Birch sap is traditionally used as a fresh refreshing drink when fresh [8], or as a fermented drink all year round [9]. It is also used in traditional medicine in many countries [10,11]. Birch sap has been known as a valuable remedy for anemia, kidney, stomach, and liver disease, arthritis, gallstones, skin diseases, gout, rheumatism and colds, infectious diseases, and intestinal parasites, as well as weakened immune systems [8,12,13]. It has also been used for hair and skincare [14]. Currently, birch sap is becoming increasingly popular as a natural probiotic that is developed by fermentation [8].

Recently, the traditional consumption of sap and its products is declining, and more and more industrially produced sap products are being used. Traditionally, sap is used fresh or fermented and consumed as kvass [1], because at the right temperature, the carbohydrates in the sap become a source of nutrients for fermenting microorganisms [15, 16]. Industrially obtained sap must be processed before the start of fermentation, because fermentation causes changes in color and odor, as well as the chemical composition of the sap [3,15].

With the increase in popularity of a healthy lifestyle, there is an increasing demand for less processed products, made while retaining as many of the original properties as possible. Up to now, the main method for preserving birch sap has been pasteurization, which reduces the health properties of the drink. Thermal treatment is still a standard sap preservation technology that can cause changes in the vitamins and other valuable nutrients, as well as a loss of original raw material quality [17,18]. Thermal technologies are also energy-intensive [19,20]. These methods are based on the indirect transfer of heat to the product using a heating medium obtained by burning fossil fuels [21]. The high energy demand causes greenhouse gas emissions, which can have a significant impact on the environment. Rising energy demand and increased global competition call for continuous improvements in technology [22].

In order for fresh sap to retain its original properties for as long as possible and to prevent fermentation, it must be processed immediately. However, this is not always possible, or it might not be economically viable to transport the product for processing in small quantities, so ways need to be found to slow down the growth of micro-organisms in sap stored under field conditions. In order to keep the product fresh and of good quality for as long as possible, alternative ways of keeping the products must be sought.

Many alternative processing methods can be used to preserve the properties of fresh birch sap apart from pasteurization: microfiltration, treatment with ultrasound, UV radiation, magnetic fields, high pressure, and various combinations of these methods. These new innovative methods inactivate the majority of microorganisms without the use of high temperatures which allows for preserving the original structure and properties of the product [3,9,15,23,24]. One such method could be the use of ozone gas. In the field of food processing, interest in ozone has grown rapidly in recent decades as consumer interest in nonthermal processing methods has increased [25,26]. Although ozone is a gaseous compound that occurs naturally in the atmosphere and is formed by lightning or high-energy ultraviolet radiation [27], both aqueous and gaseous ozone have been universally recognized as safe to be used by the food industry (GRAS—Generally Recognized as Safe) by the U.S. Food and Drug Administration (FDA) [28,29]. Researchers McClurkin et al. [29], Madanchi et al. [30], and Almeida et al. [31] confirmed that the use of ozone as an antimicrobial compound in the form of gas or as water solution is safe for food treatment and storage, drinking water, and juice treatment. According to the United States Department of Agriculture, food treated with ozone can be considered as "100% organic" or "organic" [32]. Ozonation is becoming widely accepted in the worldwide food industry as an organic technology [33]. Due to its high oxidizing capacity, ozone inactivates most microorganisms, thus prolonging the shelf life and not damaging the product [26,34,35]. Ozone treatment adds up to energy conservation because it does not require thermal energy [36]. However, studies on the ozone concentrations used in sap treatment as well as the effect of ozone on changes in sap quality indicators are scarce. The aim of this study is to determine the effectiveness of ozone at reducing microorganisms in birch sap and the effect of ozone on the composition of sap.

## 2. Materials and Methods

### 2.1. Plant Material

Location. The sap was collected in a birch stand located in a Lithuania forest (stand area: 2.4 ha; coordinates: 55°2′ N; 23°50′ E). Soil type: Luvisol; temporarily flooded mineral

soil stand parameters: age: 91 years; density: 308 trees ha$^{-1}$; average diameter at chest height: $39.4 \pm 1.2$ cm; average height: $25.0 \pm 0.9$ m; wood volume: 395 m$^3$ ha$^{-1}$.

In the spring of 2017, towards the end of sap flow season, the sap from overmatured silver birches (*Betula pendula* Roth) was collected for one day, bottled in 1-L plastic bottles, and delivered to the laboratory within 30 min at ambient temperature. The birch sap in the laboratory was immediately cooled to 2 °C.

Birch sap collection. In the forest stand, 20 silver birch (*Betula pendula* Roth) trees were selected in accordance with morphological features, followed by testing with a chemical reagent—a solution of 2,4-dinitrophenylhydrazine, which clouds up from the inner bark of the silver birch (Betula pendula Roth) and does not react with the bark of the downy birch (*Betula pubescens* Ehrh) [37].

Holes (approx. 30 cm above the soil surface) with a diameter of 22 mm were drilled in the trunks of the selected trees. A cordless drill and feather drill bit were used to drill the holes. An implant was placed at the site of the hole, which was connected by a special connection to a plastic tube. The sap was collected in a 10-L sealed plastic container. The abovementioned setup for tapping tree sap is covered by a patent (No. LT 5813 B). This ensures there are no contaminants such as insects, debris, or rainwater in the sap, and it requires no further filtration after gathering.

### 2.2. Ozonation

Preparation of samples. The collected sap was bottled in dark, sterile 1-L plastic bottles. For one test, 63 bottles were prepared: 7 bottles × 3 initial, and two storage temperatures × 3 replicates. The sap was brought directly to the laboratory from the collection site. Until preparation for ozonation, the sap was stored at 2 °C.

The methodology and the course of the research are presented in Figure 1.

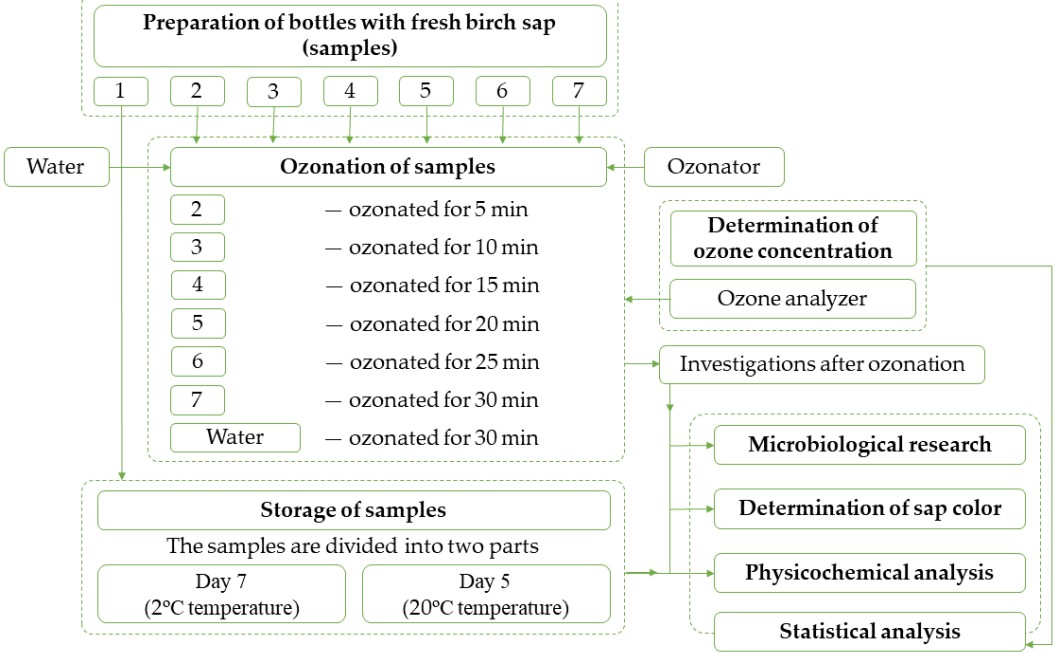

**Figure 1.** Evaluation of the influence of ozone on the biochemical composition of birch sap.

The samples were subdivided as follows: sample 1 (control)—fresh sap (non-ozonated); sample 2—sap ozonated for 5 min; sample 3—sap ozonated for 10 min; sample 4—sap ozonated for 15 min; sample 5—sap ozonated for 20 min; sample 6—sap ozonated for 25 min; sample 7—sap ozonated for 30 min.

The temperature, pH, and soluble solids content of the sap were determined before and after ozonation, and a sample was taken from each sample and saved for microbiological examination.

Ozonation of samples. A tube with an ejector, connected to the ozonator OZ-AW-15 (power: 2 kW, treated water flow: 5 $m^3$ $h^{-1}$, ozone content: 15 g $h^{-1}$) was submerged in the sap. The ozonator was switched on and ozone was fed through the tube. With the help of an ejector, ozone was evenly distributed in the sap.

Determination of ozone concentration. Sap that was not stored for further studies was tested every 5 min during ozonation to determine ozone concentration. Samples were taken from several different locations. The ozone concentration was also determined in water in parallel. To determine ozone concentrations in water and sap, the ozone analyzer "Ozone Meter Palintest" (measuring range: 0–3.0 mg $L^{-1}$) was used. The first seven samples, after ozonation and ozone concentration determination, were sent for chemical analysis.

### 2.3. Storage of Samples

After ozonation and sampling, seven specimens of sap were placed for seven days at 2.0 ± 0.1 °C and the other seven for five days at 20 ± 0.5 °C. After the storage, the temperature, pH, electrical conductivity, total soluble solids content, and microbiological indicators of sap were determined. After sampling, specimens were sent for chemical analysis. The tests were repeated three times, taking six samples from each.

### 2.4. Microbiological Research

To determine the quality of sap, a microbiological test was carried out, during which the following were determined: total microorganism count, yeasts and molds count, and the count of lactic acid bacteria. Microbiological tests were performed according to standards:

- Total microorganism count was determined according to LST EN ISO 4833-1:2013 Microbiology of the food chain—Horizontal method for the enumeration of microorganisms, Part 1: Colony count at 30 °C by the pour plate technique.
- Lactic acid bacteria count was determined according to LST ISO 15214:2009 Microbiology of food and animal feed—Horizontal method for the enumeration of mesophilic lactic acid bacteria—Colony-count technique at 30 °C.
- The total number of yeasts and molds was determined according to LST ISO 21527-1: 2008 Microbiology of food and animal feeding stuff—Horizontal method for the enumeration of yeasts and molds, Part 1: Colony count technique in products with water activity greater than 0.95.

Six replicates of samples were taken from each sample of sap for analysis. Colonies were counted; figures are represented as colony-forming units per milliliter (CFU $mL^{-1}$) [38]. To determine the changes of microorganism colony count, samples were taken from each sample immediately after ozonation as well as after storage at different temperatures, including those samples that were untreated (control).

### 2.5. Determination of Sap Color

Color coordinates in the CIE (Commission Internationale de l'Eclairage) *L\*a\*b\** uniform color space were measured with a MiniScan XE Plus spectrophotometer (Hunter Associates Laboratory, Inc., Virginia, USA). Color coordinates in the CIE *L\*a\*b\** uniform color space were measured with a MiniS-can XE Plus spectrophotometer (Hunter Associates Laboratory, Inc.) as described in [39]. Six replicates were taken from each sample.

### 2.6. Physicochemical Analysis

The research was performed in the laboratory of the Institute of Horticulture of the Lithuanian Research Centre for Agriculture and Forestry:

- Determination of total soluble solids. Total soluble solids (TSA) were determined on a PR-32 digital refractometer (Atago Co., Ltd., Tokyo, Japan). Six replicates were performed.

- Determination of titratable acidity. Titratable acidity was determined by titration of a sap sample with 0.01 N NaOH solution until the pH increased to 8.2 and expressed as a percentage of citric acid equivalent. Six replicates were performed.
- Active acidity (pH) was measured with an "inoLab pH Level 1" pH meter and a SenTix 81 (WTW) electrode (Xylem Inc., Weilheim, Germany). Six replicates were performed.
- Electrical conductivity was measured with an ECTestr 11+ conductometer (Oakton, Vernon Hills, IL, USA). Six replicates were performed.
- Determination of ascorbic acid (vitamin C) content. Ascorbic acid content was determined by titration with a 2,6-dichlorphenolindophenol sodium chloride solution [40]. In this oxidation-reduction reaction, ascorbic acid in the extract was oxidized to dehydroascorbic acid and 2,6-dichlorphenolindophenol dye was reduced to a colorless compound. The endpoint of the titration was detected when an excess of the unreduced dye gave a rose-pink color to the acid solution.
- Monosaccharides, sucrose, and total sugars were determined according to the Association of Official Analytical Chemists [41]. To investigate the effects of ozone on ascorbic acid (vitamin C), monosaccharides, sucrose, and total sugar in the sap, the highest ozonation rate was chosen. Tests were performed only on fresh and ozonated sap immediately after 30 min of ozonation. Six replicates were performed.

### 2.7. Statistical Analysis

Data from the study were analyzed using Microsoft Office Excel and IBM SPSS 20 Statistics software. One-way analysis of variance (ANOVA) and Dunnett control post hoc tests were used to assess whether the mean of one control group statistically significantly differed from the means of treatment groups. Differences between averages of treatment groups were assessed by one-way analysis of variance ANOVA with Tukey HSD (honestly significant difference) post hoc tests. Differences before and after treatment and between treatment times were considered significant at $p < 0.05$. Data were summarized as means $\pm$ standard deviations. Colony-forming unit counts CFU mL$^{-1}$ were log10 transformed. Data analysis between the chemical components of the sap was performed using correlation analysis, where the value of $p$ obtained for the evaluation of significance equated to the value of $\alpha$ (significance level: $\alpha = 0.05$).

## 3. Results

### 3.1. Ozone Concentration in Birch (Betula pendula) Sap

The dispersion of ozone gas in the sap was investigated first. For comparison, changes in ozone concentration in water (10.4 $\pm$ 0.7 °C), which had a pH of 7.67 $\pm$ 0.07 before ozonation ($n = 6$), were also monitored. According to the study data (Figure 2), it can be seen that the distribution of ozone in water is more even and easier to distribute over the entire volume of water. After 5 min, the ozone concentration in the water was 0.52 $\pm$ 0.028 mg L$^{-1}$, and no significant difference was observed between further ozonation intervals (Tukey HSD test, $p < 0.05$, $n = 6$). From 5 min onwards, the ozone concentration in water averaged out to 0.53 $\pm$ 0.01 mg L$^{-1}$. The pH of ozonated water was found to rise to 8.16 $\pm$ 0.07 ($n = 6$).

The distribution of ozone concentration in the sap was not stable throughout the ozonation period. After 5 min, the ozone concentration in the sap was 0.087 $\pm$ 0.009 mg L$^{-1}$, which is 83.3% less than in water. Fresh sap is not sterile, and ozone first reacts with microorganisms and thus decomposes faster and does not retain. The initial number of microorganisms is the highest and after most of the microorganisms are eliminated, the ozone saturates the liquid more easily. Therefore, there is a subsequent increase in ozone concentration in the sap. Later, i.e., after 10 min, it can also be seen that the ozone concentration in the sap is not as stable as in water. As is known, microorganisms are destroyed throughout the entire period of ozonation. Thus, it is assumed that the sap mixed, and the ozone reacted with the microorganisms unevenly during ozonation. It is also believed

that the unequal ozone concentration was influenced by the trace elements and other components in the sap. The highest ozone concentrations in the sap were recorded after 10 and 20 min: $1.173 \pm 0.09$ mg $L^{-1}$ and $1.25 \pm 0.08$ mg $L^{-1}$, respectively. There was no significant difference in ozone concentrations after 15, 25, and 30 min. The average ozone concentration in the sap treated with ozone for 10 min was $0.99 \pm 0.09$ mg $L^{-1}$.

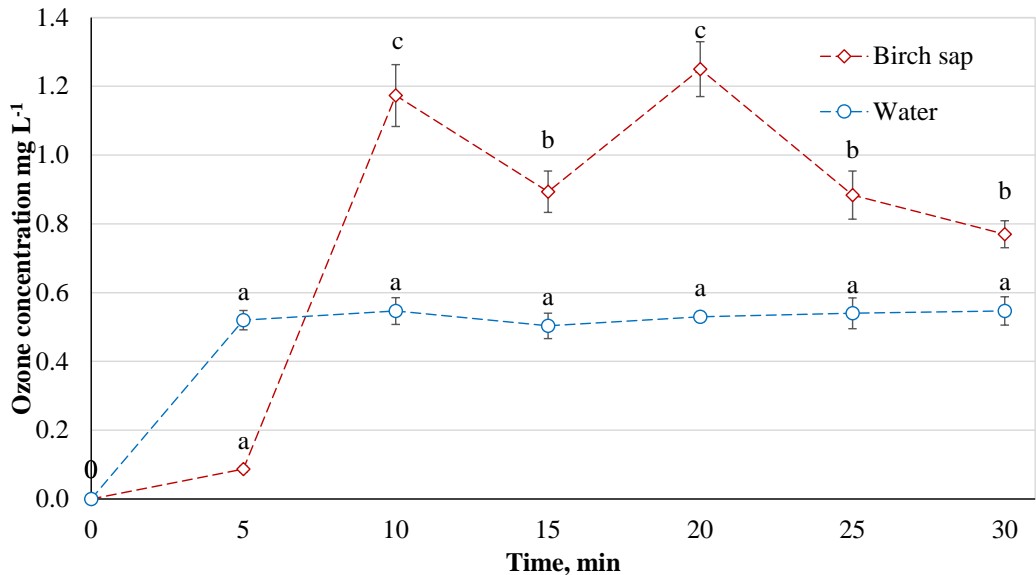

**Figure 2.** Changes of ozone concentration in sap and water. A Tukey HSD test was used to compare the means of ozone concentrations in sap and water ($p < 0.05$, $n = 6$). Means in points marked by the same lowercase letter are not significantly different ($p < 0.05$).

### 3.2. Microbiological Research

The number of microorganisms present in the sap depends on the way the sap was collected and the environmental conditions, although the sap is never naturally sterile. The initial number of microorganisms determines the further shelf life of the sap. Microbiological tests showed the total number of microorganisms (Figure 3), the number of lactic acid bacteria (Figure 4), and the total number of yeasts (Figure 5) in the initial and treated sap, after storage for five days (at 20 °C) or seven days (at 2 °C).

First, the mean of the total count of bacteria in the control samples (1) (Figure 3) was compared with the means of the total bacteria count in the other treatment groups (2–7), separately evaluating one-day, five-day, and seven-day data (Dunnett's test at a significance level of 0.05). Differences in the total bacterial count ($p < 0.05$) between sap treatment methods on different days (Tukey HSD test) were also presented.

In fresh birch sap (No. 1, day 1), the total bacterial count was found to be 6.49 log CFU $mL^{-1}$. From the data of the study, it can be observed that, in all cases, after ozonation at different intervals, the total bacterial count decreased. After a 5-min (No. 2) ozonation of the sap, the total bacterial count decreased by 67.9%; after 10 min (No. 3), when the ozone concentration on average increased to $0.99 \pm 0.09$ mg $L^{-1}$, the decrease reached 54.1%, which is 13.2% higher than after 5 min (No. 2). After 15 min (No. 4), the decrease reached 64%, and after 20 min (No. 5) it was 85.2%. There was no significant difference between these treatment intervals (Nos. 1, 2, 3, 4, 5) (Tukey HSD test, $p < 0.05$). A significant difference was found between fresh sap and intervals of treatment for 25 min (No. 6) and 30 min (No. 7), where the difference was 98.5% and 99.1%, respectively. Comparing the control group (No. 1) (Dunnett's tests, $p < 0.05$) with other treatment groups, we observed that a significant difference was recorded between the 20-min (No. 5), 25-min (No. 6), and 30-min (No. 7) treatment groups.

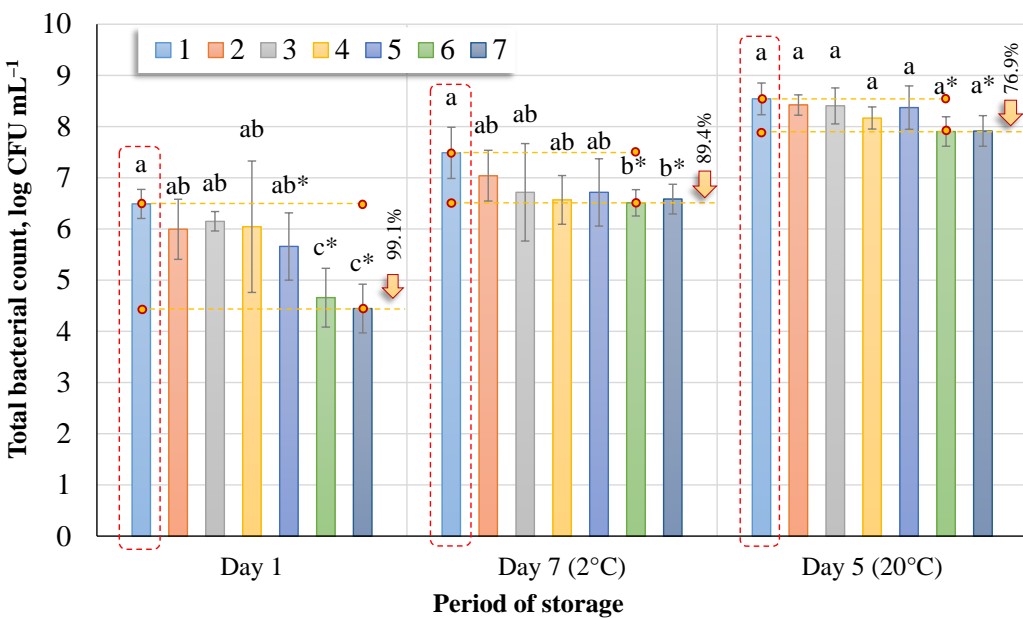

**Figure 3.** Changes in the total bacterial count in birch sap: 1 (control)—not ozonated sap (initial); 2—sap ozonated for 5 min; 3—sap ozonated for 10 min; 4—sap ozonated for 15 min; 5—sap ozonated for 20 min; 6—sap ozonated for 25 min; 7—sap ozonated for 30 min. Significant, substantial differences are distinguished by different letters (Tukey HSD test). Single letters indicate that there is no significant difference. Differences were considered significant when the *p* values were less than 0.05 (*n* = 18). In the Dunnett test, significant differences between the control group (No. 1) and the treatment group are marked with an asterisk "*".

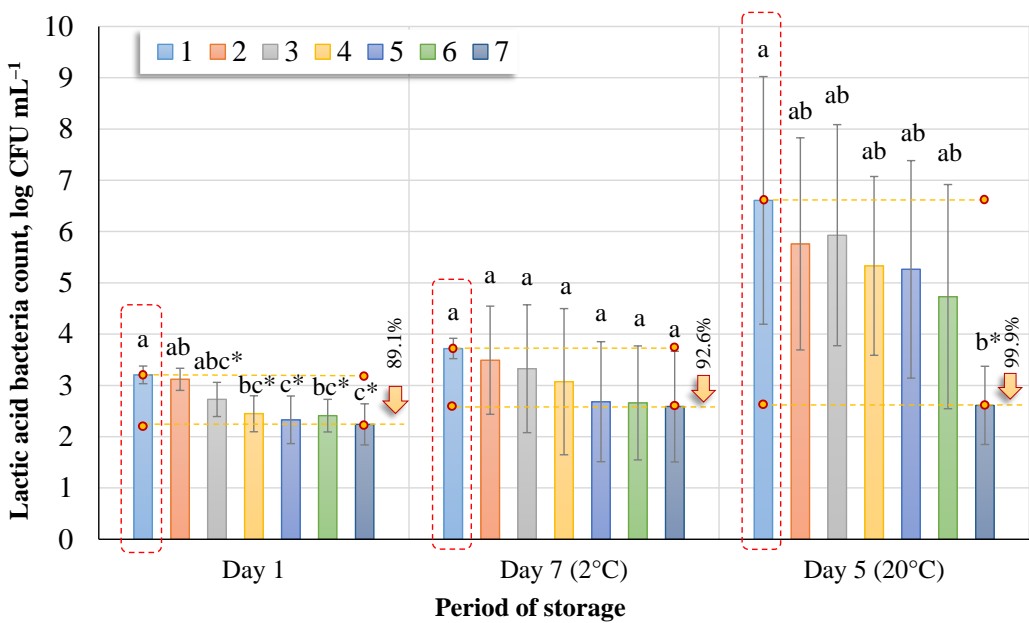

**Figure 4.** Changes in the lactic acid bacteria count in birch sap: 1 (control)—not ozonated sap (initial); 2—sap ozonated for 5 min; 3—sap ozonated for 10 min; 4—sap ozonated for 15 min; 5—sap ozonated for 20 min; 6—sap ozonated for 25 min; 7—sap ozonated for 30 min. Significant, substantial differences are distinguished by different letters (Tukey HSD test). Single letters indicate that there is no significant difference. Differences were considered significant when *p* values were less than 0.05 (*n* = 18). In the Dunnett test, significant differences between the control group (No. 1) and the treatment group are marked with an asterisk "*".

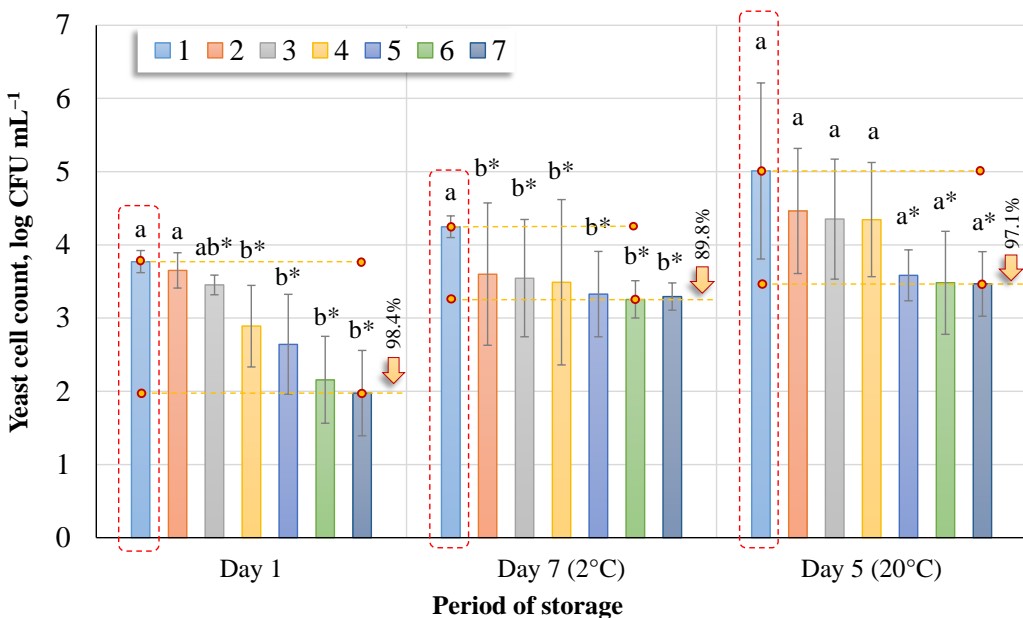

**Figure 5.** Changes in the yeast colonies count in birch sap: 1 (control)—not ozonated sap (initial); 2—sap ozonated for 5 min; 3—sap ozonated for 10 min; 4—sap ozonated for 15 min; 5—sap ozonated for 20 min; 6—sap ozonated for 25 min; 7—sap ozonated for 30 min. Significant, substantial differences are distinguished by different letters (Tukey HSD test). Single letters indicate that there is no significant difference. Differences were considered significant when *p* values were less than 0.05 (*n* = 18). In the Dunnett test, significant differences between the control group (No. 1) and the treatment group are marked with an asterisk "*".

Longer storage resulted in an increase in the total bacterial count. After seven days of storage at 2 °C, the highest recorded total average bacterial count was in the untreated sap–7.49 log CFU mL$^{-1}$ (No. 1, Day 7, 2 °C). An essential difference (Tukey HSD test and Dunnett's test, $p < 0.05$) was found between the raw sap and the treatment intervals of 25 min (No. 6) and 30 min (No. 7), where the total bacterial count was 89.4% and 87.5% lower, respectively, than in the untreated sap.

The highest total bacterial count was recorded in the untreated sap stored at 20 °C for five days. This led to storing the sap at a temperature of 20 °C for only five days, unlike the 2 °C sap that was stored for seven days. After five days of storage, the mean total bacterial count in the untreated sap was 8.54 log CFU mL$^{-1}$ (No. 1, Day 5, 20 °C). No significant difference was observed in total bacterial count in the sap (Tukey HSD test, $p < 0.05$). However, a comparison of the control group and the ozone treatment groups showed a significant difference between treatment groups 6 and 7 (Dunnett's tests, $p < 0.05$).

The total bacterial count detected in fresh sap varies depending on storage temperature (Figure 3). After seven days of storage at 2 °C, the total bacterial count increased by 13.3%, and after five days of storage at 20 °C, the bacterial count increased by 24%.

In experiments on the lactic acid bacteria count (Figure 4), it was found that the untreated sap (No. 1) contained 3.20 log CFU mL$^{-1}$ (day 1), 3.72 log CFU mL$^{-1}$ (day 7, 2 °C), or 6.61 log CFU mL$^{-1}$ (day 5, 20 °C). The lactic acid bacteria count decreased with each longer treatment interval. However, compared to the untreated control group (Dunnett's tests, $p < 0.05$), significant differences were recorded immediately after treatment in groups (No. 3, 4, 5, 6, and 7). Sap ozonation for 30 min (No. 7) had a statistically significant effect after five days when the sap was stored at 20 °C.

When comparing sap treatment methods, the most significant effect on lactic acid bacteria (Tukey HSD test, $p < 0.05$) was achieved on day 1 after ozonation of sap for 15 min, 20 min, 25 min, or 30 min, and after five days' (20 °C) storage after ozone treatment for 30 min. The lactic acid bacteria count recorded in the non-ozonated sap (Figure 4) increased by 13.8% after seven days of storage (2 °C) and by 51.5% after five days (20 °C). The most effective reduction of lactic acid bacteria count was when the sap was ozonated for 30 min.

No mold was found in the sap; only yeast was detected. Figure 5 shows the change in the number of yeasts in birch sap when the sap was tested on the first day (immediately after ozonation), after seven days of being stored at 2 °C, and after five days when the sap was stored at 20 °C.

The highest count of yeast colonies was found in untreated sap, both fresh and stored for five or seven days. The yeast colonies count in the sap decreased immediately after ozonation (day 1) from 3.77 log CFU mL$^{-1}$ (No. 1) to 1.98 log CFU mL$^{-1}$ (No. 7); after 30 min ozonation, the yeast colonies count decreased by 98.4%, while after seven days the count of yeast colonies in sap ozonated for 30 min was 88.9% lower than that in non-ozonated sap. After five days, the yeast colony count in the sap ozonated for 30 min was 97.1% lower than in the non-ozonated. Comparing the number of yeast colony-forming units (Tukey HSD test, $p < 0.05$) recorded in sap (day 1), there was no significant difference between untreated and treated for 5 min (No. 2), and between treated for 15 min, 20 min, 25 min, and 30 min. After seven days of storage at 2 °C, the number of yeast colony-forming units was lower in all ozonated samples of the sap, which was statistically significant. Data from day 5 (20 °C) showed no significant difference between the groups. Comparing the control group (Dunnett's tests, $p < 0.05$) immediately after ozonation, significant differences were found for groups 3–7 on day 7 with all ozonation groups and on day 5 for groups 5–7.

In summary, ozonation at an ozone concentration of $0.99 \pm 0.09$ mg L$^{-1}$ affected the number of microorganisms. However, it depends on the duration of ozone treatment: the longer the sap is exposed to ozone, the more effective the inactivation of microorganisms.

### 3.3. Changes in the Color of Birch Sap

When choosing the ozonation range, it is very important to take into account the qualitative indicators. As is known, during storage, if the sap is not completely frozen, there are changes in composition—that is, fermentation of the sap takes place. The higher the temperature, the faster these changes occur. At the same time, the color coordinates of the dyes change. Figures 6 and 7 show changes in birch sap hue angle (*h*) and chrome (*C*) during storage. Color is an important indicator and affects its attractiveness.

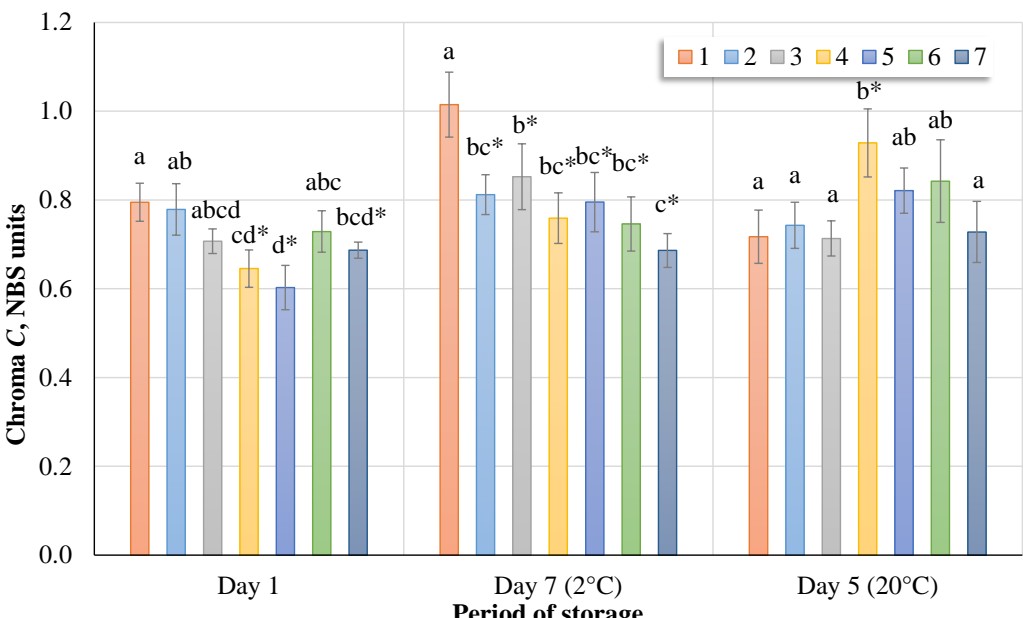

**Figure 6.** Changes in the chroma *C* of birch sap during storage: 1 (control)—untreated sap (initial); 2—sap ozonated for 5 min; 3—sap ozonated for 10 min; 4—sap ozonated for 15 min; 5—sap ozonated for 20 min; 6—sap ozonated for 25 min; 7—sap ozonated for 30 min. Dunnett tests treat one group as a control (No. 1) and compare all other groups against it. A significant difference between the control group and the other treatment groups is denoted by an asterisk "*". Tukey HSD test was used for the comparison of processing methods (marked in lowercase). The mean difference is significant at the 0.05 level (*n* = 18).

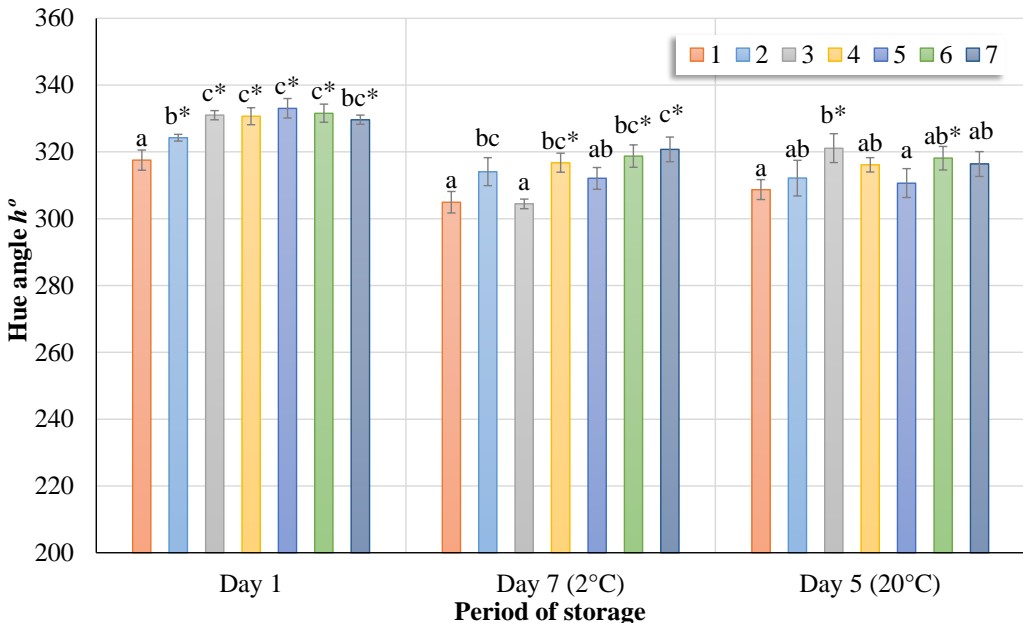

**Figure 7.** Changes in the hue angle of birch sap $h°$ during storage: 1 (control)—untreated sap (initial); 2—sap ozonated for 5 min; 3—sap ozonated for 10 min; 4—sap ozonated for 15 min; 5—sap ozonated for 20 min; 6—sap ozonated for 25 min; 7—sap ozonated for 30 min. Dunnett tests treat one group as the control (No. 1) and compare all other groups against it. A significant difference between the control group and the other treatment groups is denoted by an asterisk "*". Tukey HSD test was used for the comparison of processing methods (marked in lowercase letters). The mean difference is significant at the 0.05 level ($n$ = 18).

The $L^*$, $a^*$, and $b^*$ values of all treated sap samples did not differ significantly ($p < 0.05$) from the control. The rate of treatment did not affect the color of the birch sap. Figure 6 shows the changes in chroma of sap samples immediately after treatment, as well as sap samples stored at 2 °C and 20 °C. The chroma of the sap samples immediately after ozone treatment (Nos. 2–7) ranged from 0.78 to 0.60 NBS units, while the initial fresh sap chroma was 0.79 ± 0.04 NBS units. At 2 °C (day 7), the hue angle value of ozone-treated sap samples (Nos. 2–7) varied by 0.85–0.69 °C on average, and at 20 °C (day 5) by 0.93–0.71 °C. Untreated sap samples stored at 20 °C (No. 1) had the highest hue angle value and averaged 0.72 ± 0.06 °C.

Statistically significant differences (Tukey HSD test, $p < 0.05$) were recorded immediately after treatment between untreated sap and samples 4, 5, and 7, which were treated. After seven days, we assessed the differences between untreated sap samples and all ozone-treated samples (Nos. 2–7). After five days, we assessed the differences between treatments 1–3 and 7 and treatments 4, 5, and 6. Comparing the control group with the treatment groups (Dunnett's tests, $p < 0.05$), it was found that, immediately after ozonation, ozone had a significant effect on the chroma in sap samples treated for 15, 20, or 30 min. After seven days, a statistically significant difference between the chroma of the untreated and treated sap emerged. After five days, when the temperature was higher, a significant difference from the control group was found in group 4 (15 min).

Changes in the hue angle $h$ of birch sap are shown in Figure 7. Based on the results and a comparison of the ozone-treated samples at different intervals with the control group, the value of hue angle increased in the range of 2.1–4.7%. The difference in hue angle value between the treated samples stored for seven days and the control group is 2.3–4.9%. After five days of storage, the difference is in the range of 0.6–3.9%.

The data averages and letter differences between samples in Figure 7 show a significant difference at the 5% level of significance ($p < 0.05$) using the Tukey HSD test. The control

group (No. 1) was compared with treatments according to the Dunnett test (*) when the results are significant at the $p < 0.05$ level. Significant differences were found immediately after treatment of the samples with ozone and after storage at both temperatures. However, when visually inspecting the samples, these differences were not noticeable.

It is believed that the color of the sap would have a significant impact on the consumer's choice, as sap is known to be clear, and the sap becomes opaque at the onset of sap spoilage processes. The turbidity of the sap may be associated with the beginning of fermentation processes.

### 3.4. Physicochemical Analysis

Variation of pH and titratable acidity of birch sap. Active acidity and titratable acidity were studied in parallel during the study. The two concepts are interrelated and each of these quantities describes the quality of the product. The parameter of acidity can be used to judge the suitability of the process for processing the product. pH is important to evaluate the ability of microorganisms to be established in sap. Titratable acidity determines the total acid concentration in the sap and also better predicts the effect of acid on taste than pH.

Figure 8 shows the change of active acidity pH (A) and titratable acidity (B) of birch sap during storage. The active acidity pH (Figure 8A) of fresh birch sap was found to be about 6.20, and a pH value of 6.10–6.49 was recorded after ozonation. The titratable acidity value of fresh birch sap (Figure 8B) was 0.007 g 100 mL$^{-1}$, and immediately after ozonation, it was 0.00413–0.00817 g 100 mL$^{-1}$. According to the study data (Figure 8A), no significant differences were observed between the control group and those treated with ozone (Dunnett's test, $p < 0.05$) and between the samples (Tukey HSD test $p < 0.05$) both immediately after treatment and during storage at 2 °C and 20 °C. However, there was a visible change over time: after seven days at 2 °C the average decrease in active acidity was 8.7–20%, and after five days at 20 °C it was 12.5–31.3%. After storage for seven days at 2 °C or five days at 20 °C, the pH values dropped; the pH of fresh sap was 5.56 and 4.88, while for ozonated sap it was 5.19–5.63 and 4.45–5.34, respectively. It can be seen that at 20 °C the pH dropped faster.

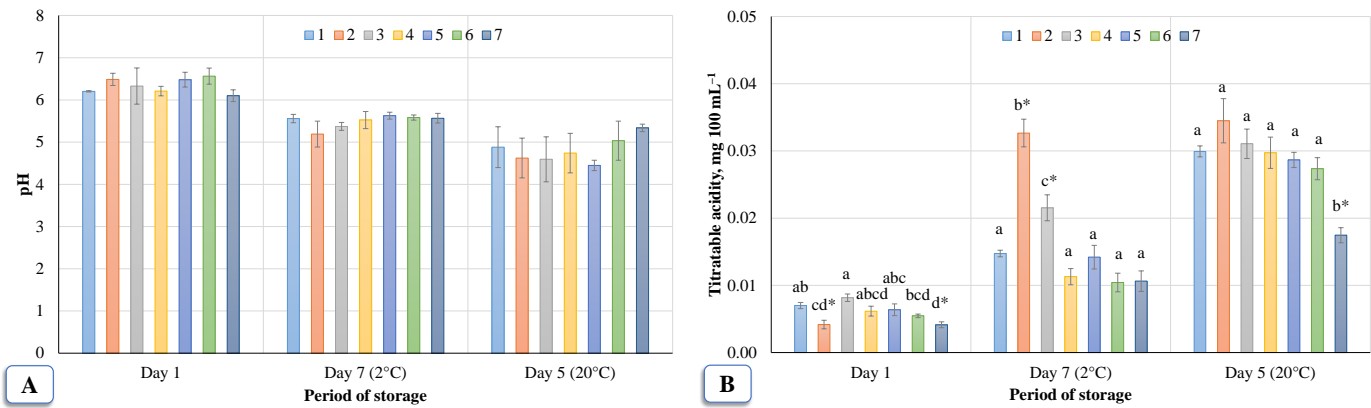

**Figure 8.** Variation of pH (**A**) and titratable acidity (**B**) of birch sap during storage: 1 (control)—non-ozonated sap (initial); 2—sap ozonated for 5 min; 3—sap ozonated for 10 min; 4—sap ozonated for 15 min; 5—sap ozonated for 20 min; 6—sap ozonated for 25 min; 7—sap ozonated for 30 min. Significant differences ($p < 0.05$) were found according to a Tukey HSD test (marked in lowercase) and Dunnett tests (marked with an asterisk "*") ($n = 6$).

Assessing the mean values of the titratable acidity (Figure 8B), it was found that, after ozonation of birch sap, the titratable acidity was significantly affected after 5 min and after 30 min of ozonation according to both a Dunnett's test compared to the control group ($p < 0.05$) and a Tukey's HSD test comparing all samples ($p < 0.05$). The acid content in these cases was 40.5% and 41.0% lower than in fresh sap. After 10 min of ozonation, a

significant (Tukey HSD test $p < 0.05$) increase in titratable acidity was observed, which was about 14%. The acidity of the other ozonated samples (4, 5, and 6) varied minimally.

After seven days of sap storage at 2 °C, a significant difference was found between the control group and groups 2 and 3 (Dunnett's test, $p < 0.05$). Compared to the Tukey HSD test ($p < 0.05$), ozonation had a significant effect on the sap, which was treated in both No. 2 and 3 ozonation rates. After five days at 20 °C, a significant difference was seen for treatment 7 when the sap was treated for 30 min (Dunnett's test and Tukey's HSD test, $p < 0.05$). From the study data, it can be observed that the titratable acidity varies greatly between days. This may be related to the fact that the acid content increases during the storage of the sap, and the rate of change depends on the storage temperature.

As mentioned earlier, fresh sap must be transparent, but when comparing the titratable acidity indicators with the color indicators, it can be seen that the color change is not yet visually identifiable with increasing acidity indicators.

Total soluble solids, electrical conductivity, and other components change in birch sap. The components accumulated in the sap are important indicators of quality. A change in these components determines the taste of the sap. Changes in birch total soluble solids and specific conductivity during storage are shown in Figure 9A,B. After evaluation by the Dunnett test at the significance level $p < 0.05$, there were no significant differences between the control group and treatment groups in terms of both sap solids and sap specific conductivity analysis.

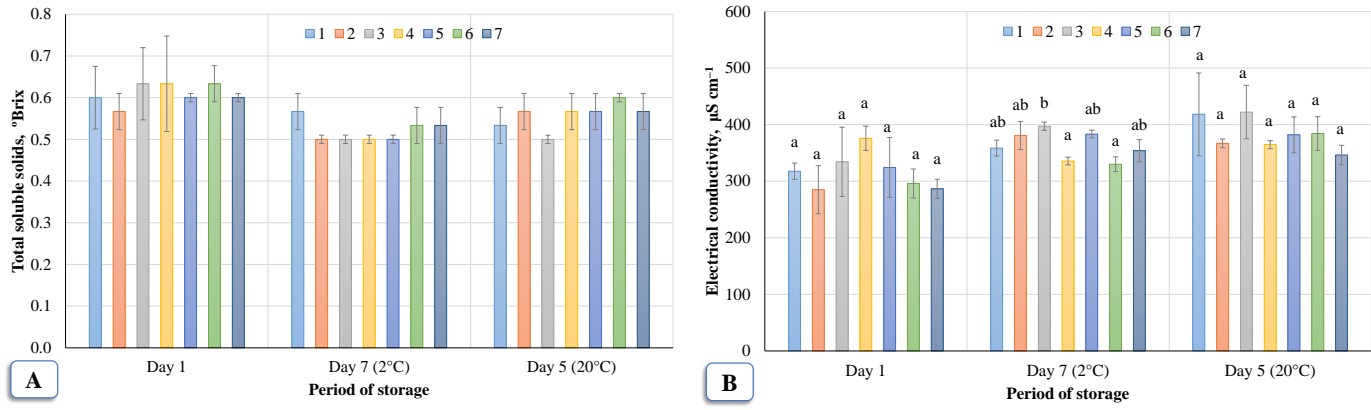

**Figure 9.** Variation of total soluble solids (**A**) and electric conductivity (**B**) of birch sap during storage: 1 (control)—not ozonated sap (initial); 2—sap ozonated for 5 min; 3—sap ozonated for 10 min; 4—sap ozonated for 15 min; 5—sap ozonated for 20 min; 6—sap ozonated for 25 min; 7—sap ozonated for 30 min. Significant differences ($p < 0.05$) were found according to a Tukey HSD test (marked in lower case) ($n = 6$).

The TSS (Total soluble solids) value of birch sap shows the percentage of dissolved substances (sucrose) in the sample (1° Brix corresponds to 1 g of sucrose per 100 g of solution). The Brix value of birch sap was 0.57–0.63° Brix. Assessing the influence of ozonation on the number of soluble solids in the sap at different temperatures (initial, 2 °C, and 20 °C) by Tukey HSD test (Figure 9A) when the significance level was $p < 0.05$ showed that ozonation had no significant effect.

Figure 9B shows the change in the electric conductivity of the sap on different days: immediately after ozonation (initial); after seven days when the sap was stored at 2 °C; and after five days when stored at 20 °C. Minimal influence of ozone was seen when the product was stored for seven days after treatment (Tukey HSD test, $p < 0.05$). Assessing the results of the sap in the following days, it was found that ozonation had no effect on the electrical conductivity of the sap immediately after ozonation or after five days of storage. To assess the correlations, a correlation analysis was performed between the acidity and the conductivity of sap (Table 1).

**Table 1.** Correlation analysis data between components in the sap and acidity values.

| | Correlation Coefficient | | |
|---|---|---|---|
| | Titratable Acidity, mg 100 mL$^{-1}$ | pH | Electrical Conductivity, µS cm$^{-1}$ |
| **Total soluble solids, °Brix** | −0.308 * | 0.283 * | −0.164 |
| **Titratable Acidity, mg 100 mL$^{-1}$** | - | −0.900 ** | 0.616 ** |
| **pH** | - | - | −0.576 ** |

\* Correlation is significant at the 0.05 level (*n* = 126). \*\* Correlation is significant at the 0.01 level (*n* = 126).

In the correlation analysis, according to Spearman's evaluation, it was observed that the relationship between pH and titratable acidity was strong (*r* = −0.9, correlation was significant at the 0.01 level). It can be seen that these indicators depend on each other. As the active acidity decreases, the titratable acidity increases, i.e., the concentration of acid in the sap increases. A medium significant correlation was found between the titratable acidity of the sap and the electrical conductivity, where the correlation value *r* was −0.576, and between the electrical conductivity and the active acidity pH, where *r* = −0.576 (correlation was significant at the 0.01 level). Figure 10 shows the components of fresh sap and their change after 30 min of ozonation when the ozone concentration in the sap averaged 0.99 ± 0.09 mg L$^{-1}$.

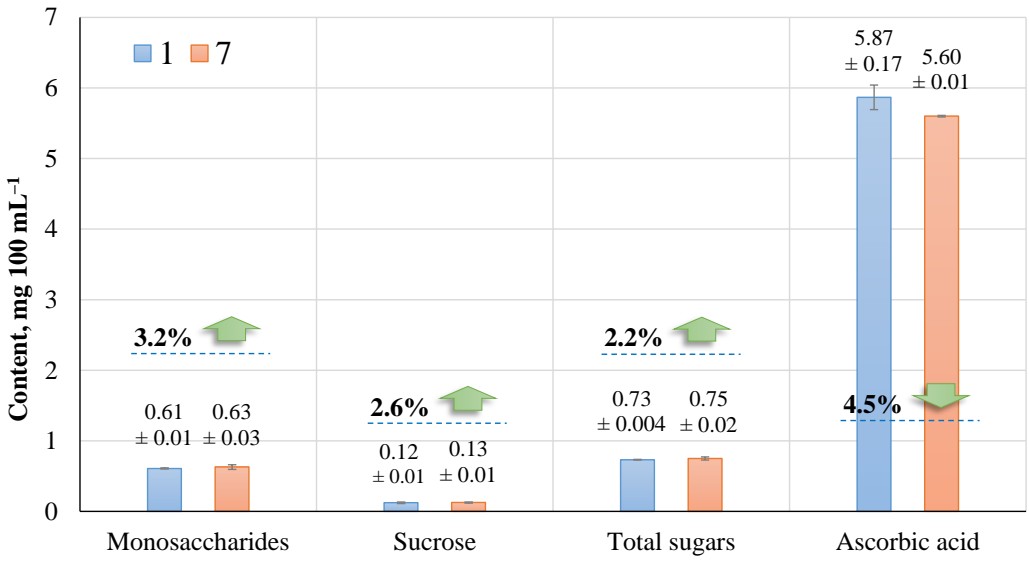

**Figure 10.** Comparison of components of fresh and ozonated for 30 min birch sap: 1 (control)—non-ozonated sap (initial); 7—sap ozonated for 30 min. Data are presented as means ± standard deviations (*p* < 0.05, *n* = 6).

According to the study, ozonation did not affect the monosaccharides, sucrose content, and total sugar present in the sap. Ascorbic acid was found to decrease by about 4.5%, but the dispersion of ascorbic acid in fresh sap was higher. It is difficult to assess the changes in sugar concentrations in sap during storage. However, a deeper study with these components should be performed as needed.

In summary, the data suggest that ozone at low concentrations is not harmful to sap or has a minimal effect. However, ozone is relatively effective in terms of the reduction of microorganisms.

## 4. Discussion

The use of ozone in the food industry is not new, but its effects on the quality of food and drink have not yet been thoroughly researched. One such well-known drink

is birch sap. A review of the studies reveals that there is still no specificity in terms of ozone concentrations and their use intervals. The aim of this study was to determine the effectiveness of ozone in the reduction of microorganisms in the sap while assessing the quality of the sap, monitoring its changes immediately after ozonation and after storing the sap in different environments. The parameters of fresh sap were compared to the parameters of ozone-treated and stored sap.

At the beginning of the study, an observation was made that ozone saturation in sap is not constant compared to ozone saturation in water. During the ozonation of the sap, the average ozone concentration reached $0.99 \pm 0.09$ mg $L^{-1}$ after 10 min. After 5 min of ozonation, only $0.087 \pm 0.009$ mg $L^{-1}$ was recorded in the sap. However, between 5 and 10 min, the ozone concentration was not studied, so it is not clear exactly at which point it rose. Compared with Labbe and Kinsley's study [42], which used similar ozone concentrations in maple sap, saturated ozone concentrations (about 1 mg $L^{-1}$) were reached within 5 min and persisted throughout the ozonation period (40 min).

The obtained results show that the number of colony-forming units of microorganisms in the ozonated sap significantly decreased due to ozone treatment. The total reduction in the number of colony-forming units of microorganisms in the sap right after ozonation was 6.32–6.48 log CFU $mL^{-1}$, the reduction in lactic acid bacteria was 2.46–3.15 log CFU $mL^{-1}$, and the reduction of yeast was 3.16–3.76 log CFU $mL^{-1}$. During the storage, ozonated sap also retained lower levels of microorganisms. Ozone's impact on microorganisms has long been known from studies with drinking water [30,43–45], various juices [18,46–49], and berries, fruit, and vegetables [50–55]. However, there were no such studies where the main objective was to reduce the number of microorganisms in birch sap using ozone treatment. One study was found on maple sap by researchers Labbe and Kinsley [42], in which the sap was treated with ozone at a concentration of approximately 1 mg $L^{-1}$. The object of the study was to make an aerobic plate count and yeast count. The researchers found that there was a significant decrease in the number of aerobic plates after 40 min, where less than 3 log were recorded, and after 20 min, the aerobic plate count decreased by only 1 log. This contradicts our study, where at $0.99 \pm 0.09$ mg $L^{-1}$ ozone concentration after 20 min of treatment there was a significant ($p < 0.05$) effect of ozone on microorganisms. As Labbe and Kinsley [42] conclude, this may have been influenced by the presence of sucrose at a concentration of 3%. In the case of the present study, the sucrose content in fresh silver birch (*Betula pendula* Roth) sap was significantly lower at $0.12 \pm 0.01$ mg 100 $mL^{-1}$. It can be concluded that the presence of organic matter affects the ozone dispersion and uniformity in the sap. It has previously been shown that the presence of organic matter, pH, or temperature can affect the effects of ozone on microorganisms [56,57]. The amount of organic matter can inhibit the effectiveness of microorganisms' inactivation. This was also confirmed by Patil et al. [18], in whose study the ozonation (75–78 µg $mL^{-1}$) effect on reducing *Escherichia coli* in orange juice started after 60 s and got deactivated in low-pulp juice after 6 min, and in unfiltered juice after 15–18 min of treatment.

More research has been carried out monitoring the changes of microbiota in the sap during storage and identifying the microorganisms as well as treating the sap with different methods by inactivating microorganisms [3,9,58,59]. In our studies, it was found that the temperature increment resulted in the growth of microorganisms in the sap. Researchers Nikolajeva and Zommere [9] found that the number of microorganisms CFU in birch sap increased rapidly during the first week of storage but the difference was not significant, regardless of the temperature (4 °C and 20 °C).

Physical, chemical, and microbiological changes occur in birch sap during storage. The rate of these changes depends on the initial parameters, storage conditions, and technology of processing. The technologies must not damage the internal parameters of the product. The composition of birch sap has been well studied. The physicochemical and rheological properties and composition of minerals, organic acids, amino acids, and sugar have been described in numerous studies [3–5,9,10,58,60,61]. Most studies on birch sap have focused

on yield and composition, while, to our knowledge, no studies have reported the effects of ozone on birch sap constituents, pH, and color.

Birch sap is colorless and, according to the results of this work, ozone had an effect on both chroma and hue angle parameters. It can be concluded that ozone is a factor that reduces the color of the sap. However, a visual difference in the color of birch sap is difficult to discern. There are no exact data on how ozone affects the color of birch sap. Some researchers have reported that the ozonation of fruit and berry juices—orange [62,63], melon [64], grape [65], apple [66], and blackberry [67]—changes the color. There are several studies in which ozone did not induce a color change: by treating prebiotic orange juice with ozone at a concentration of 0.057, 0.128, or 0.230 mgO$_3$ mL$^{-1}$ of juice (15, 30, 45, or 60 s) [31] or by treating apple juice with 0.08–0.31 mg L$^{-1}$ ozone at different temperatures [68]. According to Sung et al. [68], the color change of juice in different studies may be due to different treatment methods and control parameters (concentration, ozone gas flow, and treatment interval).

During the study, it was found that the values of the active acidity of fresh birch sap and immediately after ozonation ranged from 6.10 to 6.49. During storage, the values of pH decreased after seven days at 2 °C and after five days at 20 °C. A larger decrease was seen after seven days when the sap was stored at 20 °C. The pH values obtained for fresh sap are in most cases consistent with other studies. Kallio and Ahtonen [61] found that the pH of birch sap can vary from 7.5 to 5.5 in the spring. The study data showed that the pH dropped faster at 20 °C. It was confirmed by Jeong and Jeong et al. [3] that pH drops during storage, and Nikolajeva and Zommere [9], by studying the physicochemical properties of the sap, found that the pH decreased faster at 20 °C (pH: 3.41) compared to 4 °C (pH: 3.79) after 58 days. The pH of fresh birch sap studied by Kallio et al. [60] ranged between 6.0 and 6.6 and dropped only 0.2–0.3 in pH units during storage. According to Kallio et al. [69], in published studies of birch sap, the pH ranges from 5.5 to 8.

The titratable acidity depended on the pH, i.e., the titratable acidity increased with decreasing active acidity. The titratable acidity increased during storage. Reviewing the values recorded in other studies, the titratable acidity in birch and maple sap was 0.50 and 0.70 mmol NaOH per liter of sap, respectively [5]. In our study, the TSS value of birch sap was about 0.57–0.63° Brix. This is lower than observed in another study where birch (*Betula pendula* Roth and *B. pubescens* Ehrh.) sap contained about 0.7° Brix [60]. Kallio et al. [69] reported that the average soluble dry matter content of birch sap was 0.5–1.8° Brix. However, this may be due to differences in birch species and the area from which the sap was collected. The electrical conductivity of fresh birch sap increased during storage. This was confirmed by Nikolajeva and Zommere's [9] study of birch sap. Another study found a significantly higher specific conductivity, which contradicts our research. According to Kūka et al. [5], the electrical conductivity of Latvian birch (*Betula pendula* Roth.) and maple (*Acer platanoides* L.) sap is 629 and 967 S cm$^{-1}$, respectively, which characterizes the total mineral content as well as the acidity of the sap. It is generally considered that fresh birch sap intended for consumption can be stored for up to two months (0–2 °C). Although the titratable acidity and pH are most likely to change during storage at this temperature, the soluble solids content and the electrical conductivity are the least likely to change [70].

Comparing the physicochemical parameters of birch sap found in our study, in several cases, it was close to the results of other studies performed. However, it was confirmed that the sugar content and other parameters of the sap depend on the type of birch, habitat, and meteorological conditions [10,71]. According to studies in the Russian Far East, these carbohydrates reach 0.9% at the beginning and end of the sap flow and rise to 1.3% in the middle [72]. Zajączkowska et al. [73] found that the sugar content of silver birch sap ranged from 0.25% to 2.25%. According to Kallio et al. [74], who studied the composition of the sap of *Betula pendula* Roth, *Betula pubescens* Ehrh, and *Betula pendula* var. *carelica* Mercklin trees in Finland, glucose (2.5–4.7 g L$^{-1}$), fructose (2.3–4.5 g L$^{-1}$), sucrose (<0.7 g L$^{-1}$), galactose (<0.05 g L$^{-1}$), sugar alcohol myoinositol (trace), malic acid (0.1–0.7 g L$^{-1}$), succinic acid (<0.1 g L$^{-1}$), citric acid (<0.1 g L$^{-1}$), phosphoric acid (<0.04 g L$^{-1}$), and fumaric acid (traces)

were detected. In southeastern Poland, *Betula pendula* Roth sap was found to contain a significantly higher sugar content: $0.46 \pm 1.03\%$ of carbohydrates, of which glucose is $0.93 \pm 0.39\%$, fructose is $1.21 \pm 0.49\%$, and sucrose is $0.32 \pm 0.24\%$ [75]. Sap (*Betula pendula* Roth. and *B. pubescens* Ehrh.), which is used for syrup production in Finland, had a maximum glucose and fructose concentration in late April or early May of 5–8 g L$^{-1}$ [76]. Kūka et al. [5] showed that silver birch sap contained mainly fructose ($5.39 \pm 0.05$ g 100g$^{-1}$), glucose ($4.46 \pm 0.04$ g l00 g$^{-1}$), and sucrose ($0.58 \pm 0.01$ g 100 g$^{-1}$), and the average ascorbic acid concentration was only 3.2 mg L$^{-1}$. In Lithuania, birch sap contains on average 1.85% sugars, of which monosaccharides (1.46%) and sucrose (0.39%) are present [77].

There are no studies on the effects of ozone on the physicochemical parameters of birch sap. However, compared to changes in the parameters in other juices when exposed to ozone, in many cases, it is consistent with our results that ozone has minimal or no effect. For example, Tiwari et al. [62] reported that ozonation of freshly squeezed orange juice—applying ozone at a concentration of $0.6-10.0\%$ $w/w$ for 0–10 min—did not result in any significant changes in pH, °Brix, titratable acidity, or turbidity ($p < 0.05$). Almeida et al. [31] did not find a negative effect of ozone on the phenol content, antioxidant capacity, or pH of orange juice. Sung et al. [68] used ozone ($<0.4$ mg L$^{-1}$) with heat in apple juice studies and found that it had no effect on quality. According to Porto et al. [78], ozone had no effect on the pH, soluble solids, acidity, and color parameters of coconut water. However, there are studies in which the changes were larger than expected, but in those cases, the ozone concentrations were significantly higher. Significant changes in apple juice quality parameters (vitamin C, carotenoids, and total antioxidant activity) were observed when using gaseous ozone at a concentration of $7.0 \pm 2.4$ g L$^{-1}$ for 30 and 60 min, but the total phenol content was significantly increased in ozonized juice [64].

In summary, this study provides initial evidence that ozone, at low concentrations, is effective at reducing bacterial, yeast, and lactic acid bacteria concentrations in birch sap. This should encourage the industry to further investigate the commercial application of the technology. However, the introduction of this technology into production requires a number of technical issues to be addressed, such as the ozone concentration and treatment interval used, the ozone depletion rate, the dynamics of ozone depletion, possible toxic by-products, and energy consumption and efficiency. These issues will be addressed in further research.

## 5. Conclusions

Rational use of ozonation on birch sap can reduce the number of microorganisms in the sap. For ozonation of sap using ozone with a concentration of $0.99 \pm 0.09$ mg L$^{-1}$, the total reduction in bacterial colony count was up to 6.32–6.48 log CFU mL$^{-1}$, for lactic acid bacteria up to 2.46–3.15 log CFU mL$^{-1}$ and for yeast up to 3.16–3.76 log CFU mL$^{-1}$. The most significant effect of ozone was after ozonating the sap for 25 and 30 min. After seven (at 2 °C) or five (at 20 °C) days of storage, the numbers of microorganisms in the ozonated samples were observed to experience a statistically significant decrease compared to the control. Ozone was found to affect the chroma and hue angle of birch sap, although this difference was not visually noticeable. Ozonation was found to have no significant effect on pH, titratable acidity, and °Brix values. After evaluating the effect of ozonation for 30 min on the monosaccharides, sucrose, total sugars, and ascorbic acid in the juice, it was found that these values varied within the margin of error.

## 6. Patents

For birch sap extraction, a tapping device (Patent No. LT 5813 B) was used. Inventors: Vladas Vilimas, Raimondas Vilimas. Vilnius, 2012-02-27.

**Author Contributions:** Conceived and designed the study, J.M., V.V., E.B., and A.R.; investigation, S.P., J.M. and P.V.; writing—original draft preparation, S.P; writing—review and editing, S.P., J.M. and P.V. All authors have read and agreed to the published version of the manuscript.

**Funding:** This study was funded by the Vytautas Magnus University Agriculture Academy and by the Institute of Horticulture, Lithuanian Research Centre for Agriculture and Forestry. The work is partly attributed to the EUREKA Network Project E! 13496 "OHMDRINKS" (No. 01.2.2-MITA-K-702-08-003).

**Conflicts of Interest:** The authors declare no conflict of interest.

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
