# Peer review of "Influence of Ozone on the Biochemical Composition of Birch Sap"

_applsci, doi:10.3390/app11072965_

Round 1

Reviewer 1 Report

Ozone is widely used as disinfectant agent in the food industry. Detailed information related to the effects of ozone treatment on the quality and microbial parameters of birch sap has not been available yet.  Manuscript applsci-115199 focuses on biochemical composition of birch sap exposed to ozone treatment with different ozonation time.

The manuscript is general well written with a logic structure. Introduction section is a good summary ot he background of the research; research motivations are clearly defined. Materials and methods are given in details. Authors investigated in details the change of chemical, microbial and colour parameters. Manuscript contains interesting and valuable results. Results can be considered as relevant and utilizable at industry scale, as ell..

Comments, suggestions:

Authors do not provide detailed explanations why was not stable the ozone saturation (Page 13, line 485-492)

Have the authors information about the sensory properties (taste, etc) of ozonated sap?

Have the authors information about the energy efficiency and costs of ozonation process?

Author Response

Response to respected Reviewer

 First of all we would like to thank the Reviewer for additional comments, detailed recommendations and contributions to improving the quality of this manuscript “Influence of ozone on the biochemical composition of birch sap”. All of the issues raised in the Reviewer comments were corrected or comment in more detail and are listed below. All corrections and answers are highlighted in the attached document.

REVIEWER 1

Comments and Suggestions for Authors

Ozone is widely used as disinfectant agent in the food industry. Detailed information related to the effects of ozone treatment on the quality and microbial parameters of birch sap has not been available yet.  Manuscript applsci-115199 focuses on biochemical composition of birch sap exposed to ozone treatment with different ozonation time.

The manuscript is general well written with a logic structure. Introduction section is a good summary ot he background of the research; research motivations are clearly defined. Materials and methods are given in details. Authors investigated in details the change of chemical, microbial and colour parameters. Manuscript contains interesting and valuable results. Results can be considered as relevant and utilizable at industry scale, as ell..

Comments, suggestions:

Comment 1:

Authors do not provide detailed explanations why was not stable the ozone saturation (Page 13, line 485-492)

Answer 1:

Following the reviewer's remark, a more detailed explanation on uneven ozone saturation in sap was supplemented in Section 3.1. Ozone concentration in birch (Betula pendula) sap:

„... Fresh sap is not sterile, and ozone first reacts with microorganisms and thus de-composes faster and does not retain. The initial amount of microorganisms is the high-est and after most of the microorganisms are eliminated, the ozone saturates the liquid more easily. Therefore, there is a subsequent increase in ozone concentration in the sap. Later, i.e. after 10 min, it can also be seen that the ozone concentration in the sap is not as stable as in water. As is known, microorganisms are destroyed throughout the entire period of ozonation. Thus, it is assumed that the sap mixed and the ozone re-acted with the microorganisms unevenly during ozonation. It is also believed that the unequal ozone concentration was influenced by the trace elements and other components in the sap. This proves that the concentration of ozone in a liquid and its dispersion depends on the pH and purity of the liquid. ...“

Comment 2:

Have the authors information about the sensory properties (taste, etc) of ozonated sap?

Answer 2:

As far as our expertise allowed, we observed the sap during the study and took notes of the observations. As we did not notice any significant changes during the study, we did not perform individual sensory tests according to the methodology (taste, smell, etc.). We also did not do so due to the lack of funding at the time, as such a study would have had to be commissioned from another institution. In our opinion, a biochemical study would be sufficient, as the taste properties are determined by the components in the sap. However, following the reviewers' comment, the results section has been supplemented for greater clarity:

„.... However, when visually inspecting the samples, these differences were not noticeable.

It is believed that the color of the sap would have a significant impact on the consumer’s choice, as sap is known to be clear and the sap becomes opaque at the onset of sap spoilage processes. The turbidity of the sap may be associated with the beginning of fermentation processes.

This may be related to the fact that the acid content increases during storage of the sap, and the rate of change depends on the storage temperature.

As mentioned earlier, fresh sap must be transparent, but when comparing the titratable acidity indicators with the color indicators, it can be seen that the color change is not yet visually identifiable with increasing acidity indicators. ..."

Comment 3:

Have the authors information about the energy efficiency and costs of ozonation process?

Answer 3:

The aim of the study was not to estimate the energy efficiency of sap ozonation. However, this area is very interesting for our team, in the near future it is planned to cooperate with manufacturers and conduct research at the production facility, where larger quantities are processed. Birch sap, unlike many fruit juices, does not contain many natural preservatives – acids and sugars, which increase the efficiency of pasteurization, so pasteurization high temperature pasteurization requires a temperature of at least 85 ℃ as well as a long exposure time – about 15 minutes (based on practical experience in sap production). which requires a powerful heater of at least 2 – 3 kilowatts for even a small amount of sap (30l). A 0.2 – 0.5 kilowatt ozone generator is enough to ozonate the same amount of sap for 10-15 minutes, that is, the ozonation cost would be about 10 times lower than pasteurization at high temperatures.

Reviewer 2 Report

The overall quality of the manuscript is very good. The experimental design is sound and the discussion of results is thorough. I only have a few comments:

1) Lines 72-74: Can the authors comment a little more on these methods of sap processing?

2) Lines 128-131: A small schematic of the treatment setup would be really helpful. Does not need to be detailed.

3) Results Section 3.4: Can the authors elaborate more on whether the changes in pH and visual quality mentioned (Line 398) would have an effect on the properties as well as ultimately the consumer perception?

4) What would be the amount of Ozone (and the power consumption required to produce it) to perhaps use it in a sap processing plant? 

Author Response

Response to respected Reviewer

First of all we would like to thank the Reviewer for additional comments, detailed recommendations and contributions to improving the quality of this manuscript “Influence of ozone on the biochemical composition of birch sap”. All of the issues raised in the Reviewer comments were corrected or comment in more detail and are listed below. All corrections and answers are highlighted in the attached document.

REVIEWER 2

Comments and Suggestions for Authors

The overall quality of the manuscript is very good. The experimental design is sound and the discussion of results is thorough. I only have a few comments:

Comment 1:

1) Lines 72-74: Can the authors comment a little more on these methods of sap processing?

Answer 1:

Following the reviewer's note, the introduction to the methods used to process the sap was supplemented by:

„... Many alternative processing methods can be used to preserve the properties of fresh birch sap apart from pasteurization: microfiltration, treatment with ultrasound, UV radiation, magnetic fields, high pressure, and various combinations of these methods. These new innovative methods inactivate majority of microorganisms with-out the use of high temperatures which allows to preserve the original structure and properties of the product. [3,9,15,23,24]. One such method could be the use of ozone gas. In the field of food processing, interest in ozone has grown rapidly in recent decades as consumer interest in nonthermal processing methods has increased [25,26]. ...“

Comment 2:

2) Lines 128-131: A small schematic of the treatment setup would be really helpful. Does not need to be detailed.

Answer 2:

In the light of the reviewer's remark and for better clarity, Figure 1 is provided:

The methodology and the course of the research are presented in Figure 1.

Figure 1. Evaluation of the influence of ozone on the biochemical composition of birch sap.

…”

Comment 3:

3) Results Section 3.4: Can the authors elaborate more on whether the changes in pH and visual quality mentioned (Line 398) would have an effect on the properties as well as ultimately the consumer perception?

Answer 3:

As far as our expertise allowed, we observed the sap during the study and took notes of the observations. As we did not notice any significant changes during the study, we did not perform individual sensory tests according to the methodology (taste, smell, etc.). We also did not do so due to the lack of funding at the time, as such a study would have had to be commissioned from another institution. In our opinion, a biochemical study would be sufficient, as the taste properties are determined by the components in the sap. However, following the reviewers' comment, the results section has been supplemented for greater clarity:

„.... However, when visually inspecting the samples, these differences were not noticeable.

It is believed that the color of the sap would have a significant impact on the consumer's choice, as sap is known to be clear and the sap becomes opaque at the onset of sap spoilage processes. The turbidity of the sap may be associated with the beginning of fermentation processes.

This may be related to the fact that the acid content increases during the storage of the sap, and the rate of change depends on the storage temperature.

As mentioned earlier, fresh sap must be transparent, but when comparing the titratable acidity indicators with the color indicators, it can be seen that the color change is not yet visually identifiable with increasing acidity indicators."

Comment 4:

4) What would be the amount of Ozone (and the power consumption required to produce it) to perhaps use it in a sap processing plant?

Answer 4:

For industrial use, the ozone content of the sap should be about 0.99 ± 0.09 mg L-1.

As this area is of great interest to our team, in the near future it is planned to cooperate with manufacturers and conduct research at the production facility, where larger quantities are processed. Also, there are plans of performing life cycle analysis in parallel. Laboratory tests are more expensive. It should be adapted to the specific case, according to the quantities of the sap, the need, etc.
